# Risk Assessment Methods in Mining Industry— A Systematic Review

**Agnieszka Tubis** [1] , **Sylwia Werbińska-Wojciechowska** [1,*] **and Adam Wroblewski** [2]

[1] Faculty of Mechanical Engineering, Wroclaw University of Science and Technology, 50-370 Wroclaw, Poland; agnieszka.tubis@pwr.edu.pl

[2] Faculty of GeoEngineering Mining and Geology, Wroclaw University of Science and Technology, 50-421 Wroclaw, Poland; adam.wroblewski@pwr.edu.pl

[*] Correspondence: sylwia.werbinska@pwr.edu.pl; Tel.: +48-71-320-34-27



**Featured Application: This article is focused on a literature review in order to provide a valuable resource for understanding the latest developments in risk management and assessment in the mining sector. The conducted research will be useful for many people, including risk managers, mining engineers, and researchers, who are interested in risk management/engineering issues. The authors believe that the conducted literature review will introduce the readers to the major up-to-date theory and practice in risk management/assessment problems in the mining sector. The presented study gives the possibility to identify the thematic structure related to risk assessment/management for the analyzed industry sector. In addition, it shows which topics from the studied scientific area are the most investigated in a given country/region. At the same time, the conducted analysis gave an opportunity to develop future research directions in the areas identified as research and knowledge gaps.**

**Abstract:** Recently, there has been a growing interest in the mining industry in issues related to risk assessment and management, which is confirmed by a significant number of publications and reports devoted to these problems. However, theoretical and application studies have indicated that risk in mining should be analyzed not only in the human factor aspect, but also in strategic (environmental impact) and operational ones. However, there is a lack of research on systematic literature reviews and surveys of studies that would focus on these identified risk aspects simultaneously. Therefore, the purpose of this article is to develop a literature review in the area of analysis, assessment and risk management in the mining sector, published in the last decade and based on the concept of a human engineering system. Following this, a systematic search was performed with the use of Primo multi-search tool following Preferred Reporting Items for Systematic Reviews and Meta-Analyses (PRISMA) guidelines. The main inclusion criteria were: (a) not older than 10 years, (b) article written in English, (c) publication type (scientific article, book, book chapter), (d) published in chosen electronic collections (Springer, Taylor and Francis, Elsevier, Science Direct, JSTOR). This resulted in the selection of the 94 most relevant papers in the area. First, the general bibliometric analysis was conducted. Later, the selected papers in this review were categorized into four groups and the critical review was developed. One of the main advantages of this study is that the results are obtained from different scientific sources/databases thanks to using a multi-search tool. Moreover, the authors identified the main research gaps in the area of the implementation of risk management in the mining industry.

**Keywords:** risk assessment; mining industry; hazard event; disruptions

## 1. Introduction

　　Mining has always constituted one of the most dangerous industries. This is confirmed by data published in Eurostat, OECD (Organisation for Economic Co-operation and Development), or by national organizations, such as, in Poland, the State Mining Authority. The reports presented by these organizations indicate the main risk groups and the effects of their occurrence in mining plants. Prepared reports on accidents in mining indicate their causes and circumstances of occurrence. Thanks to this, it is possible to develop standards relating to actions taken to improve health and safety at work in mining, public safety and environmental protection [1].

　　Moreover, the importance attributed to the risks associated with mining operations is determined not only by the fact that it is one of the most dangerous sectors of the economy, but also by the scale of mining operations. Figure 1 shows total mining productions by continents in tons.

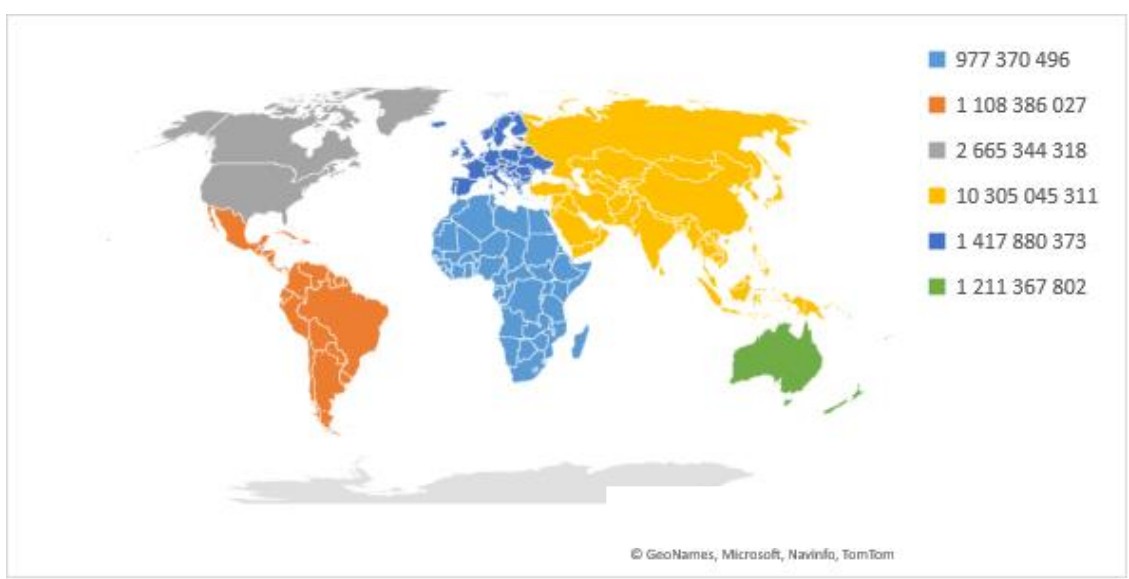

**Figure 1.** Total mining productions by continents in 2018 in tons (developed on the basis of data available in the World Mining Data database. (World Mining Data provides an indispensable basis for commodity forecasts and activities in minerals policy at national and European level; it contains production of mineral commodities listed in detail by continents, country groups, development status, per capita income, economic blocks, political stability of producing countries, largest producers and others. The data are available online: https://www.world-mining-data.info/ (accessed on 13 July 2020).

　　Such an intensive mining process, which results in a huge scale of production, generates many risks related to both the operations and resources used, but also to the interaction between the mining system (mines) and the environment. This makes research on risk analysis, assessment and management for this sector particularly important, especially regarding ecological, social and economic aspects. Therefore, the demand for research in this area and new publications, especially for the most productive areas, such as Asia and North America, should continue to grow.

　　Because mines are a complex human engineering system, they are exposed to multi-faceted risk. Often, the result of this risk occurrence is the loss of life and health of people. It is important to note that these effects may apply not only to employees of mines, but also to the environment—i.e., for example, residents of areas adjacent to the mine. For this reason, the mining sector has been focusing for several years on the need to implement and develop various risk assessment and management concepts. This risk should be analyzed not only in the professional aspect (human factor) but also in strategic (environmental impact) and operational aspects (safety of machines and devices, correctness of the implemented mining process). Research conducted in Polish mining enterprises for several years confirmed that the attention of managers has been focused primarily on the specific risk that

comes from within the mining company and is associated with the occurrence of natural and technical hazards, the effects of which are particularly severe for human health and life [2].

The emphasis on implementing the concept of risk management in mine operations is also reflected in the laws, regulations and standards appearing in subsequent years that relate to risk assessment and management. An analysis of the currently applicable standards in this area has allowed us to distinguish 19 documents dedicated to the mining sector. It is worth noting that these standards can also be classified in accordance with the above-mentioned division of the analyzed risk into general and human-, machine-, and environment-focused standards. The division of the analyzed standards and directives is presented in Figure 2. Part of the presented standards are on a global scale, while others are related to the region of the European Union, but the figure also shows documents that are valid only in Poland. A detailed description of the standards is given in Appendix A, Table A1.

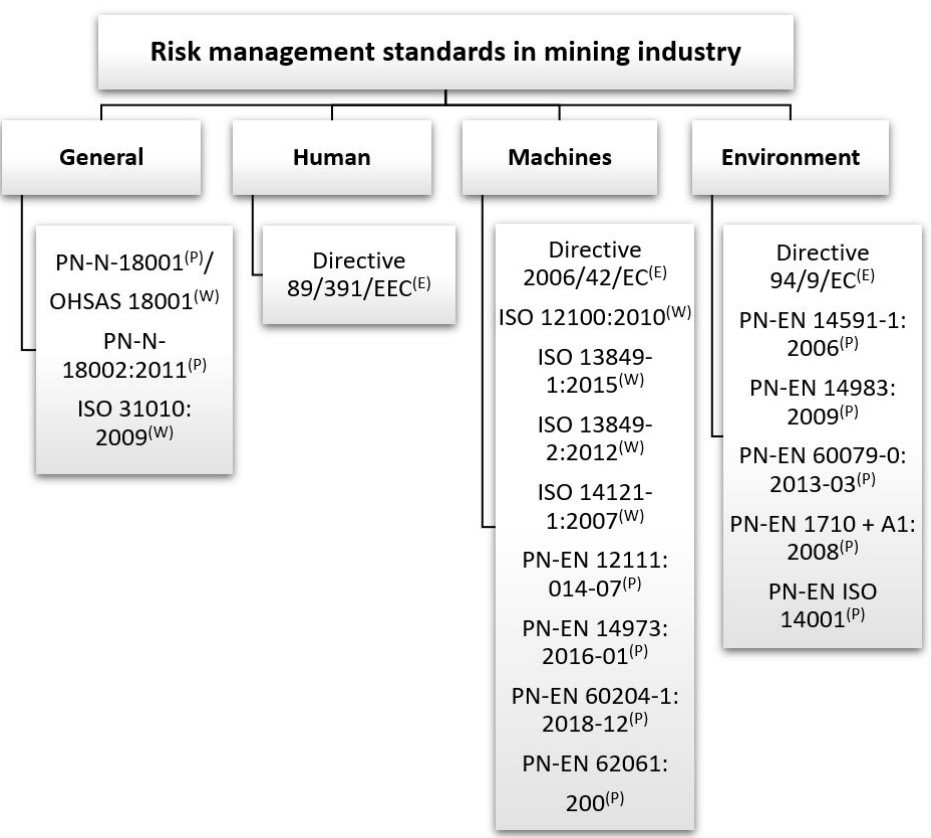

**Figure 2.** Classification of the main risk-related standards for the mining sector (where: (W)—global scope of application, (E)—standards applicable in Europe, (P)—standards applicable in Poland).

The increasing importance of risk management in mining processes is also indicated by commercial reports prepared for the purpose of managing the mining sector. One such report is the *Mining Risk Review*, which is published by Willis Towers Watson. This report appeared for the first time in 2014, and since 2016 it has been published periodically every year. Each report deals with a different topic related to risk, but they all focus on emerging challenges for the mining sector and the threats therein. The list of topics covered in the years 2016–2019 is presented in Table 1. The second periodical risk report that deserves attention is the *Risk and Opportunities for Mining* that has been appearing for two years, published by KPMG International [3,4]. These reports present the results of research on state of mining industry—risks and opportunities, key trends, and managers' expectations for their organizations.

**Table 1.** The subjects of reports *Mining Risk Review*.

| Year | Title | Goal | Main Topics |
|------|-------|------|-------------|
| 2016 | Mining Risk Review 2016. Dealing with uncertainty [5] | Highlighting key developments within the industry and focusing on risk management issues | Private equity capital; social license to Operate; advances in 3D printing; maintaining tailings dam; geotechnical, people and environmental risk |
| 2017 | Mining Risk Review 2017. The future of mining is now [6] | Determination of four key challenges that mining industry must address in new, innovative ways and focusing on risk mitigation and transfer issues | Geopolitics; stakeholder relations; digitization; people |
| 2018 | Mining Risk Review 2018. Six key messages for the mining industry today [7] | Determining six key messages that are critical in ensuring that the industry remains on track | Mining risk is no longer an option; greater attention for managing project delivery; avoiding a regulatory headache; geopolitical tensions as a significant threat to the industry; Global insurance market capacity as a threat for thermal coal risks; possible change in insurance market dynamics |
| 2019 | Mining Risk Review 2019. Addressing uncertainty [8] | Addressing the uncertainties of mining risk and mining risk transfer | Digitization; bottlenecks; geopolitical risk; social economic development |

The growing interest within the mining industry in issues related to risk assessment and management is also reflected in conducted scientific research. Therefore, in recent years, there have been more and more publications devoted to these issues. As a consequence, a large number of articles in a given area results in the appearance of review articles aimed at gathering, structuring and classifying knowledge about published scientific results. Analysis of publications from the last decade regarding literature reviews in the area of risk in the mining sector has allowed us to distinguish 20 articles. As well as standards, these articles can be thematically qualified into four groups: general, human factor, machine, and environment. The largest number of review articles concern research on the environmental impact of the mining sector [9–16]. Comparable attention was paid by researchers to conduct reviews on the risks related to the human factor. For this area, six review papers have appeared in the last decade. Analyses of human factor research have focused primarily on issues related to human health and safety (including accidents) [17–20] and work organization and team management [21,22]. There is a visible lack of review articles in the area of risks associated with mining machinery. The analyses carried out allowed us to identify only two reviews of literature devoted to machinery, while taking into account the human factor issues [23,24]. The remaining four review articles were classified as general as they did not concern any of the groups distinguished above and were more general in nature [25–28].

Therefore, the aim of the article is to develop a literature review in the area of analysis, assessment and risk management in the mining sector, including: (1) biometric analysis of publications from the period 2010–2020 using the Preferred Reporting Items for Systematic Reviews and Meta-Analyses (PRISMA) method, and (2) thematic analysis of the scope of analyzed publications, aimed at grouping research according to the adopted classification (human-machine-environment). Following this, the main contributions of this paper include:

- a summary of the research developed in the mining sector in the last decade in the area of risk assessment, risk management, risk analysis, and risk decision,
- conducting the qualification procedure in accordance with the adopted distribution criteria based on the concept of functioning of human engineering systems in the mining sector,
- identification of research gaps in the area of implementation of risk management concepts in the mining sector.

In conclusion, the outline of this review paper is as follows: in Section 2, we explain the method used to select and scan relevant journal articles on the topic of risk in mining industry, which conforms to the PRISMA guidelines. This section also describes the strategy used for literature search process performance and criteria that were applied to assess the relevance of analyzed documents. Section 3 describes the main results of conducted bibliometric analysis. Section 4 is focused on the presentation of results of thematic analysis aimed at grouping research according to the defined classification. Later, in Section 5, the literature research and knowledge gaps are identified. Finally, Section 6 ends with the concluding remarks and recommendations for future studies.

## 2. Review Methodology

The presented systematic review was conducted based on the PRISMA guidelines, given in [29]. The chosen method gives the possibility to properly search and select relevant scientific literature on the given topic with defining research objectives and providing clear quantification of scientific developments in a specific field of knowledge [30–33]. Following this, this section explains the document search and selection process with the definition of eligibility criteria and identification of relevant papers for further investigation.

### 2.1. Literature Search Strategy

The literature searching process was based on the use of multi-search tool Primo (Primo is a scientific search engine which enables the simultaneous searching of many information resources, including databases, magazine and e-book services of various publishers and suppliers, contents of library catalogues, as well as other digital sources; available online: http://biblioteka.pwr.edu.pl/e-zasoby/wyszukiwarka-primo (17 June 2020)), which gave the possibility to analyze many information resources, including, among others, ScienceDirect database, Elsevier and Springer publishers' databases, or the JSTOR database. The literature search was conducted between 8 June 2020 and 14 June 2020.

Primo is a scientific search engine that allows for the simultaneous searching of many information resources, which are searched in a quick and easy way, using a single search window, and the search results are displayed on a single platform in the form of a consolidated list of results. One of the basic functions of the PRIMO tool is filtering and narrowing the results. Therefore, during the selection process, the authors used the offered functionality of the chosen multi-search tool. However, the selection criteria were determined based on the authors' experience.

The initial searching procedure was based on the following search term "*risk in mining industry*". The first step of the searching procedure gave the possibility to identify 208,814 relevant records. In the next step, in order to focus on relatively new applications, problems and technologies, the searches were limited to studies published during the last 10 years. Additionally, only documents written in English were considered. Based on these exclusion criteria, 101,903 records were identified.

Later, the authors focused on filtering studies, taking into account four inclusion criteria— the results were limited to those publications which contained one of the following phrases in the title: risk analysis, risk assessment, risk management, and risk decision. These criteria were established based on expert experience (two authors) and reflect the most relevant aspects of the analyzed research area. As a result, the screening process had the purpose of filtering out papers that were not related with the main topic. Thus, the identified records were scanned by title. Out of the initial 101,903 records, 100,034 were eliminated during the screening process. Moreover, the search was limited only to the following types of documents: journal articles, books, and book chapters. These were selected according to the reason that they would be likely to present a good variety of unique approaches in the analyzed research area. The last selection criterion regarded the type of online databases used. The searching procedure was limited to such online databases as Springer (all available), Taylor and Francis, Elsevier with ScienceDirect, and JSTOR. This choice was dictated by the fact that these are the most important online databases with full access availability for authors. After applying these rejection criteria, the documents were reduced to 742. Moreover, 6 publications were deleted as duplicates.

In result, there were defined 736 papers, which were later fully read in order to identify the most relevant papers.

## 2.2. Selection Process

The obtained documents were subsequently examined by two independent reviewers (A.T. and S.W). The main goal of this step was to verify which of the articles were potentially eligible to be used for a further qualitative and quantitative analysis. The main criterion applied in the full text analysis was its relevance to the investigated thematic area and defined groups. The studies that described risk issues in other industries were excluded. Moreover, studies were excluded from further analysis that examined such problems as product development strategy or general issues potentially applicable in mining industry (but not confirmed).

After a consensus between the authors of this systematic review, 642 papers were rejected. They were consensually considered as being out of scope after reviewing the full document. Consequently, a total of 94 manuscripts were included for a further qualitative and quantitative analysis. Figure 3 represents the flow diagram of the selection of studies according to PRISMA statements.

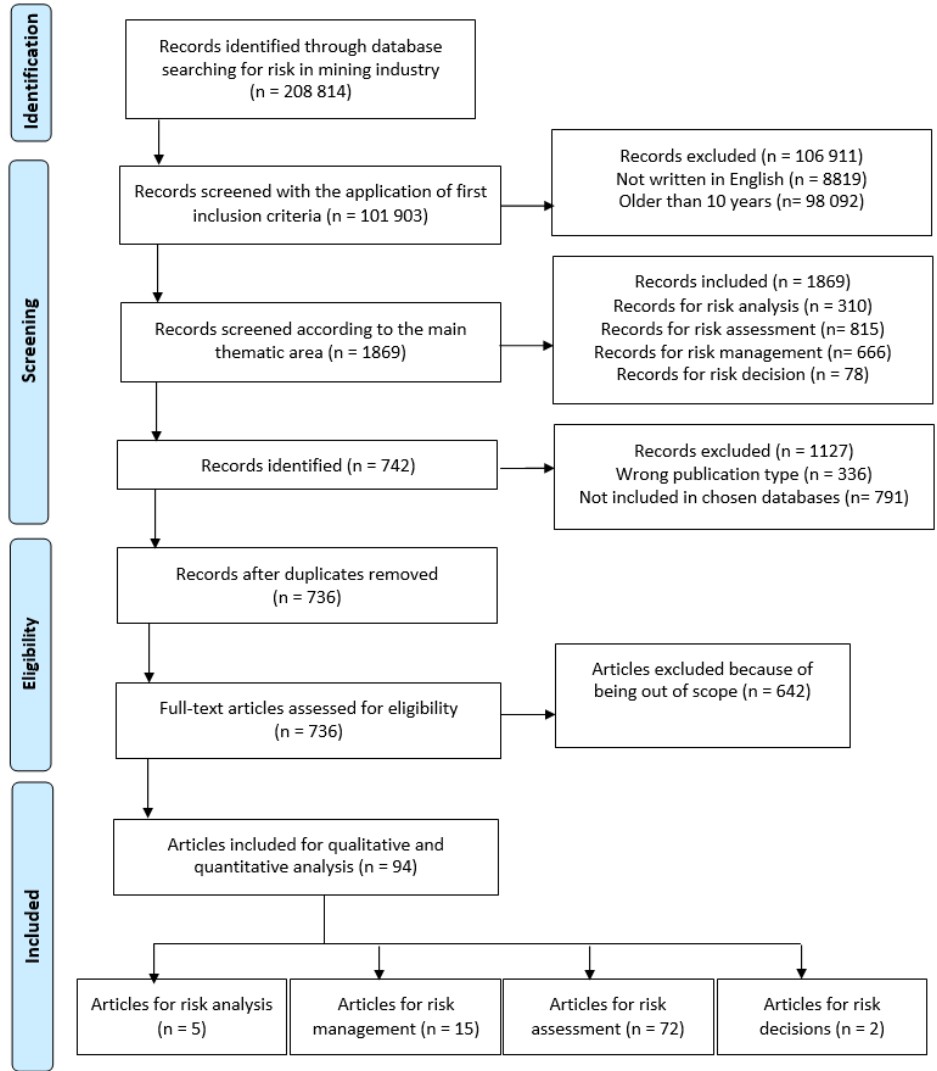

**Figure 3.** Preferred Reporting Items for Systematic Reviews and Meta-Analyses (PRISMA)-based flowchart of the systematic selection of the relevant studies in the analyzed research area.

## 3. Results

At this stage, a detailed bibliometric analysis was carried out for the selected articles from four thematic groups for risk in mining industry from the last decade.

Ninety-four articles from four analyzed areas were adopted for detailed analysis. Most publications were found for the keywords "*risk assessment*", which together accounted for almost 80% of all the analyzed texts. The number of publications for each of the analyzed search terms was:

- Two publications in the area of *risk decisions*,
- Five publications in the area of *risk analysis*,
- Fifteen publications in the area of *risk management*,
- Seventy-two publications in the area of *risk assessment*.

The analysis of the authors' and scientific centers' origins allows us to state that most of the publications from the studied area come from China (32 articles), Australia (10 articles), the USA and Canada (7 articles each). The regions of origin of the authors of the analyzed publications are shown in Figure 4.

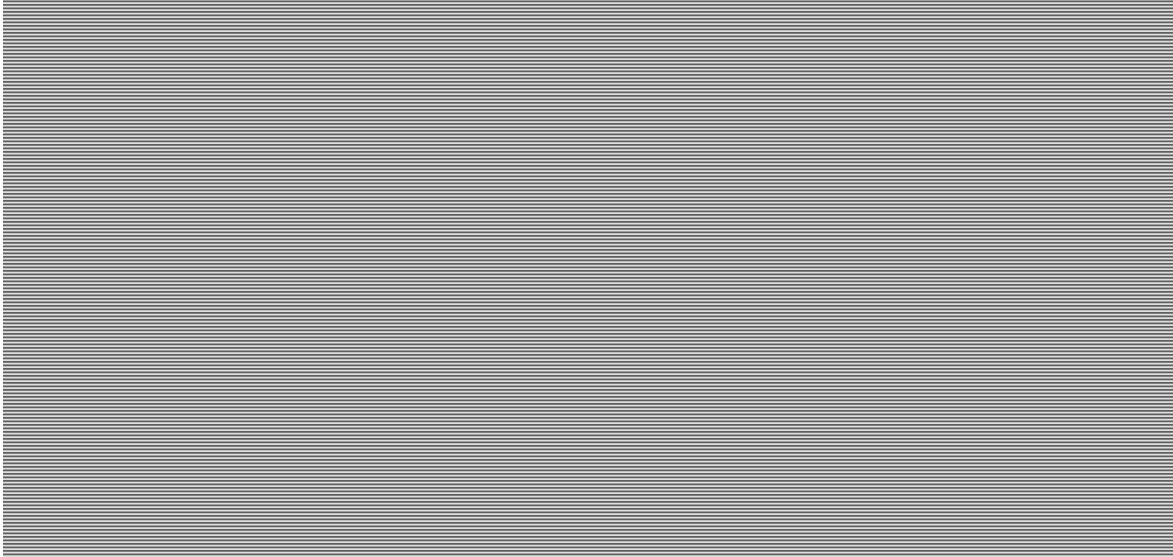

**Figure 4.** Number of papers by location where the study took place.

The analyzed publications were limited in step 2 of the adopted methodology to those published during the last decade. The adopted limitation seems to be correct, as the verification of the years in which subsequent articles were published indicates a clearly growing trend from 2015. As shown in Figure 5, for the last five years, the annual number of publications has been above 10, while in previous years it did not exceed six articles per year. This suggests that the topic risk assessment and management in the mining sector is far from being exhausted, and its popularity among researchers is still rising. It is safe to say that further developments and unique studies regarding this field of knowledge will keep appearing in the near future.

Articles concerning risks in the mining sector have appeared in many studies. The 94 publications under analysis have been published in a total of 45 journals, of which more than 70% include one article each. A detailed list of journals in which the analyzed research results were published is shown in Figure 6. The figure shows that only those scientific journals for which at least two articles from the analyzed 94 were identified. The largest number of publications appeared in the journal *Human and Ecological Risk Assessment* (12 articles). Numerous publications can also be found in *Environmental Geochemistry and Health* and *Environmental Monitoring and Assessment* (nine articles in each journal). Such a large number of publications in these top three journals is mainly due to the fact that research

on risk in the mining sector refers to its environmental impact. This is confirmed by the analysis of the review articles presented in Section 1, as well as by the list of thematic areas presented in Section 4. It should be noted, however, that the topic of risk in the mining sector is of interest not only to journals devoted to the mining sector or environmental issues. Some publications also appeared in safety journals (*Journal of Safety Research, Safety Science, Food Security, Journal of Loss Prevention in the Process Industries*), as well as those related to management and production research (*Production Planning and Control, Journal of Cleaner Production*). The detailed analysis that also contains the presentation of investigated papers according to the place of their publication is given in Appendix B, Table A2.

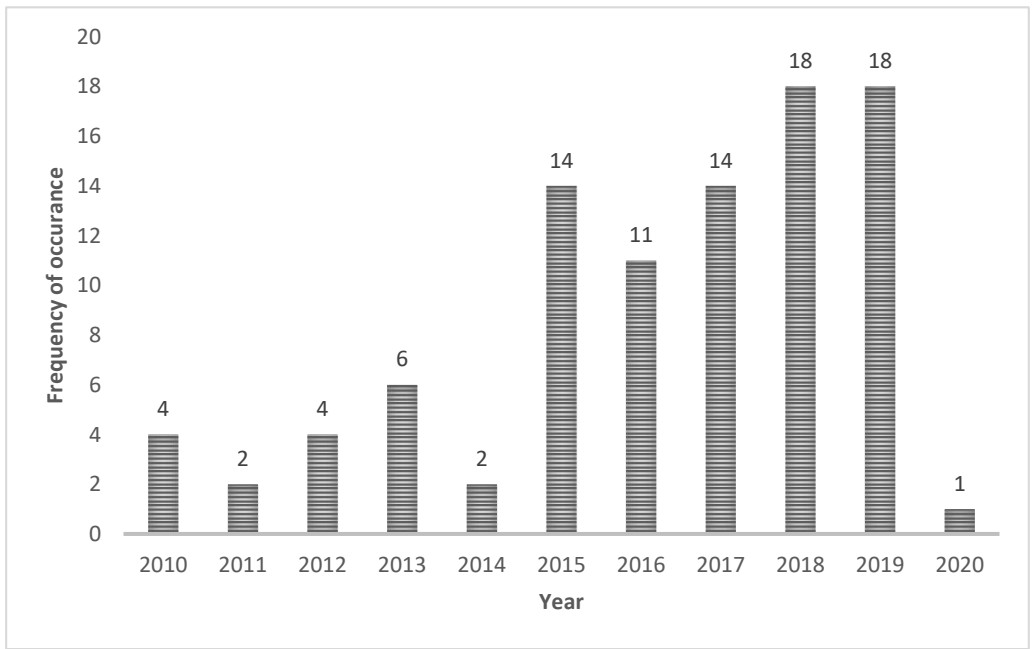

**Figure 5.** Number of papers by publication year.

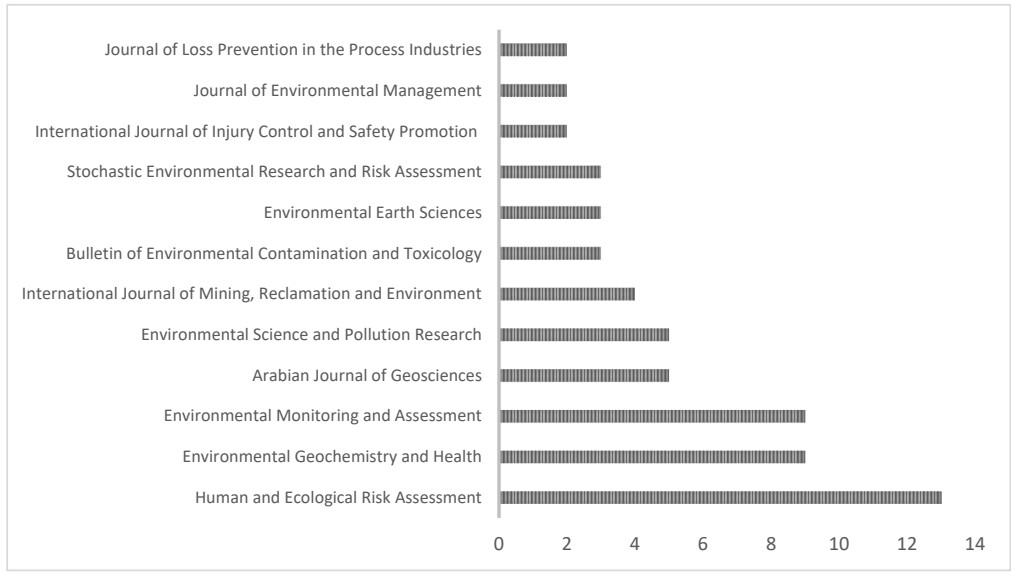

**Figure 6.** Number of papers in each investigated journal (for journals with at least two published papers from the analyzed 94 articles).

The conducted biometric analysis also concerned the repeatability of the indicated keywords used in the papers (Figure 7). The highest share among repeated keywords has the phrase *risk assessment*,

which was indicated in 34 articles. It is significant that the term *heavy metal(s)* is placed in the second place. If we combine this result with the number of repetitions of this concept, only in the single number—*heavy metal(s)*—the concept of this keyword occurred in 26 of 94 articles. This indicates the main area of research, to which the publications on risk in the mining sector are devoted. Since the conducted research is largely focused on the negative impact of the work of mines on the environment, *mining/mining activity* is in third place regarding the most frequently used keywords.

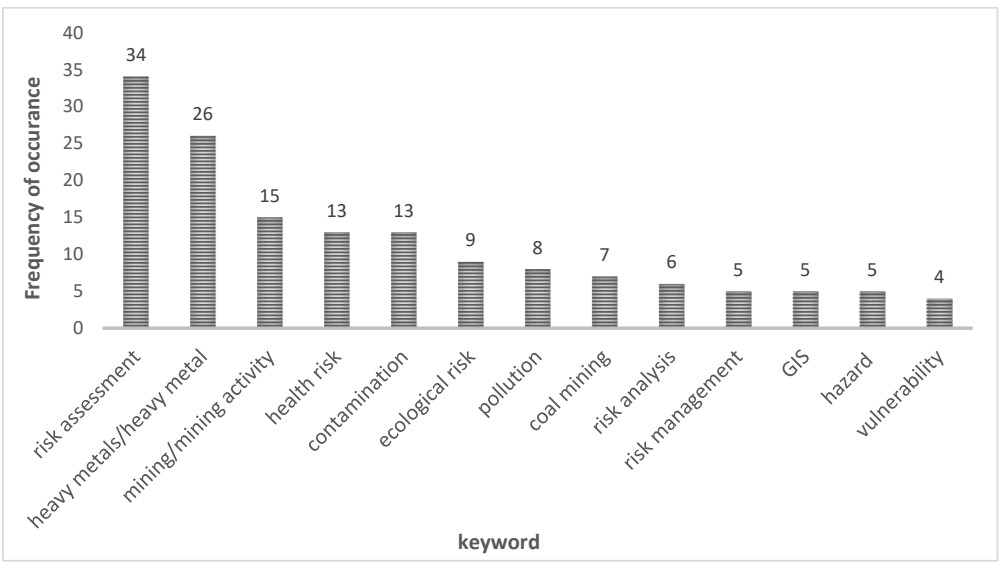

**Figure 7.** Number of most frequently occurring keywords in the analyzed papers.

## 4. Thematic Analysis of the Conducted Review

This section provides a detailed analysis of the selected papers. In order to clearly present the main thematic areas that are covered by the identified papers, the mind map was developed (Figure 8).

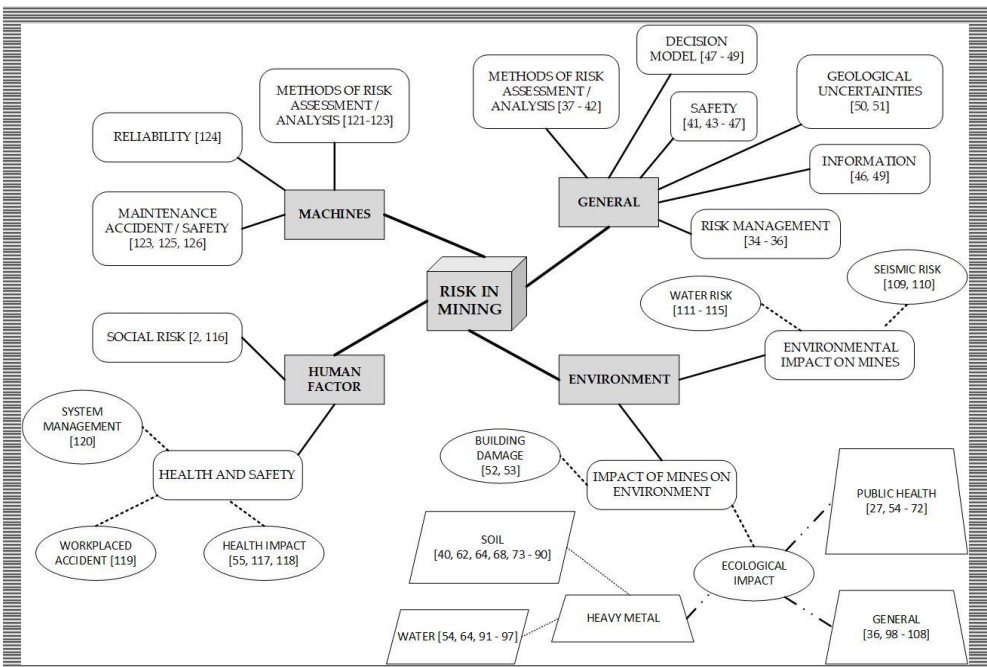

**Figure 8.** The main problems analyzed in the selected papers.

According to the given Figure 8, the main classification based on the four main defined previously groups: human factor, machines, environment, and general. In each of these groups, there were defined the main problems analyzed in the selected papers. The four indicated basic thematic groups are marked in grey. For these groups, characteristic research areas were distinguished, which are repeated in the analyzed articles. For the group *machines* and *general*, one level of conceptual branches was defined. For example, three thematic subgroups were defined, namely *maintenance accident/safety*, *reliability*, and *methods of risk assessment/analysis,* for the thematic group *machines*. Additionally, for the group *human factor*, there were defined two main subgroups, and in the area of *health and safety*, a second level of conceptual branches was defined. For the last thematic group, *environment*, there were defined up to four conceptual levels. Issues assigned to each of the distinguished conceptual levels were marked with a different shape and different linear connections in order to increase the readiness of the developed map.

Furthermore, it should be noted that several of the investigated articles should have been classified in more than one thematic area due to their complexity. Therefore, they were presented on the mind map in all the areas in which they should be classified due to their thematic scope. In addition, in Appendix B, Table A2, their multidisciplinary character was indicated in the column of thematic areas.

First, the group that encompasses general issues was analyzed. In this group, six main research problems were identified. A few papers were dedicated to risk assessment (RA) and risk management (RM) issues. In the RM area, papers focused on the problems of mining project risk management [34], or procurement and contract management of construction services [35]. The model for the joint implementation of risk assessment and risk management was presented in [36], where two case studies were provided. Another approach for risk management was given in [37], where the authors, based on the ISO 31000:2009 standard, used and proposed a tool for complex system investigations.

Implementation of Monte Carlo (MC) simulations for risk assessment performance was presented in [38,39]. The first paper was focused on the development of stochastic simulation to quantify uncertainty in mineral deposits and allowing better management of the geological risk during mining scheduling. The second one was focused on the comparison of different correlation approaches on risk analysis associated with uncertain parameters of mining ventures in order to uncover which one would yield the most accurate result.

The other research problems in this area regarded risk assessment of agricultural soil contamination [40], safety violations in underground bituminous mines [41], and assessment of the risk of roof falls [42].

Safety issues in the mining sector were under consideration in six papers. The problems investigated in this area regarded multi-method or multi-criteria analyses approach implementation (e.g., [43,44]) or decision-making modelling use (e.g., [45–47]). In work [43], safety leadership analysis was performed, adopting a multi-method approach, in which the critical decision method, Rasmussen's risk management framework and the Accimap method were applied. The fuzzy-VIKOR-based approach for safety and risk assessment in the mining industry was presented in [44]. The decision-based modelling approaches were related to issues such as safety measure system development in underground coal mines [45], information uncertainty [46], or the evaluation of safety of coal mining above a confined aquifer [47].

Decision-making problems were also analyzed in works [48,49]. The authors in [48] introduced a method using a multi-goal fuzzy cognitive map (FCM) and multi-criteria decision making based on sensitivity analysis to assess the risks associated with working accidents in underground collieries. The second work [49] was focused on the characteristics of concepts and methods for evaluating sequential information gathering schemes in spatial decision situations.

Moreover, in the analyzed thematic group, the problem of geological uncertainty was identified. In work [50], the authors proposed a new systematic framework to quantify the risk of kriging-based mining projects due to the geological uncertainties. The second interesting view of this problem was

given in [51], where the authors provided guidelines for designing and organizing a geotechnical risk assessment process to satisfy the underground mining needs.

The second analyzed group is focused on the environmental issues and is the most represented. The relationships between the environment and the mines are bilateral. On the one hand, the mine, while carrying out its mining activities, directly influences the environment and generates certain negative effects on the environment in the form of, e.g., pollution of the air, water, and soil, health problems, and mining damages. The identified papers in this area were focused on two main issues: building damages and ecological problems, and were assigned to the group *impact of mines on environment*. On the other hand, however, the environment may also have a negative impact on the activities of the mine through natural processes that interfere with its proper operational performance, such as earthquakes or flooding with groundwater. The articles related to the risk assessment for this area were classified in the second group—*environmental impact on mines*.

Building damages were under the investigation of the authors of work [52]. They presented an approach to building damage risk assessment in mining induced areas, which is based on a comparison between buildings strength and terrain deformation. In another work [53], the authors proposed an integrated system comprising deep mining, coal-gangue dressing, and underground backfill mining. They also developed many numerical models for buildings aimed at studying the surface subsidence and deformation.

The second area mostly regarded public health issues (see works [27,54–72]) and the impact of heavy metals on the environment in the area where the mine operates (see works [40,62,64,68,73–90] for soil contamination investigation and works [54,64,91–97] for water contamination problems). In this group, there is also identified the general sub-group, where issues such as, among others, sustainability [98], environmental resource management [99,100], mining dilution [101], acid rock drainage [102,103], landscape scale risk assessment [104], or ecological risk assessment [105–108] were analyzed.

In the environmental thematic group can be defined the second approach, where the environmental impact on mining operation performance was investigated. In this area, research works were focused on seismic risk assessment (e.g., [109,110]) and water risk assessment (water inrush problems [111–113], water resources in mining [114], and surface water contamination [115]).

Another thematic group regards human factor issues. In this group, two main research areas were identified, namely social risk and health and safety.

In work [2], the problem of environmental and social risk management during the process of colliery liquidation was considered. The proposed conception is based on including the sustainable development and corporate social responsibility in the total system of risk management in a mining enterprise. In the second work [116], the authors introduced an approach for addressing social risk in mine feasibility studies.

Health and safety issues were analyzed by researchers in three aspects. The problems of health consequences of residents regarded disease risk assessment problems [117], radiation risk [118], and odor and dust influence on community [55]. Moreover, there was one reported work devoted to workplace accidents [119], and one work for system management [120].

The last thematic group was focused on machines. In this area, three main problems were investigated. The first regarded risk assessment method implementation in the mining industry for their machines and the proper operation of processes. For example, in work [121], the authors recognized risks in terms of operation, safety, geology, environment, finance, maintenance, repair, reliability, offer, and availability, and used preventative and mitigated controls to eliminate/reduce risks. In [122], this problem was analyzed with the use of a fuzzy logic-based safety evaluation method. However, in work [123], the authors proposed a method for risk assessment of mining machines, taking into account reliability and maintenance constraints. The issue on reliability was also analyzed in work [124]. The authors developed a methodology for risk assessment in order to support maintenance management based on criticality analysis, root cause analysis, and a tool for generation of

effective and efficient solutions (TRIZ) implementation. Moreover, the maintenance problems were also investigated in [125], where the authors discussed the hazards connected to maintenance and operability of the equipment with the customer and identify safety improvements. The OHS/safety information management in the Australian coal industry was investigated in [126].

The articles classified in accordance with the distinguished four thematic groups were analyzed in relation to the countries in which the publications from a given area had appeared. The obtained results of the conducted analysis are presented in Figure 9.

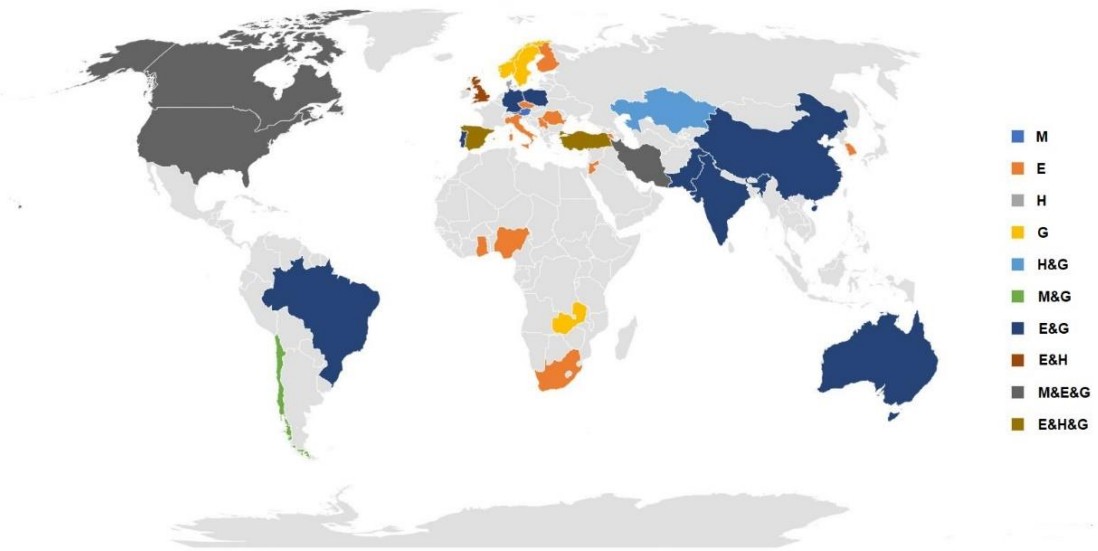

**Figure 9.** The thematic areas of publications appearing in a given region (where: M—machine; E—environment; H—human factor; G—general).

The analysis of the results presented in Figure 9 shows that in many countries, research is most often conducted in the areas of *environment* and *general* issues. These topics dominate in Australia, parts of Asia (mainly China and India) and Brazil. In North America (USA and Canada), publications from the area of *machines* are additionally included in this group. In Chile, on the other hand, there are only general publications on the risks associated with *machines*. It is also worth noting that the topics in individual articles from Africa are mainly related to the *environment* thematic group.

## 5. Identification of the Main Research and Knowledge Gaps

Based on the conducted literature review, the main research and knowledge gaps in the area of risk management in mining sector can be defined.

Mining operators are complex human engineering systems, where the source of the risks involved may be man, machine or the system environment. Therefore, for the needs of the analysis, three research areas have been identified, which indicate the element of the mining system to which the risk analysis relates (*machines, human factor, environment*) and an additional general group (*general*). The largest share in the assessed material was held by research devoted to risk assessment in the area of environment. It should be noted, however, that the research described concerned both the impact of mining systems on the environment and the impact of the environment on mine operations.

The analysis shows that this research area can be considered as a knowledge gap for the area of risk assessment and management in the mining sector. This is the result of the bilateral relationship described in the Section 4, which links these two subsystems (environment and mining company). So many publications in this area are also the result of current trends in sustainable development concept implementation. The mining industry is a significant contributor to the environment and its immediate surroundings. Therefore, there is an understandable need for research results from this

area. The literature review also suggests that this issue has not yet been fully explored and described. Therefore, it can be expected that the risk aspects of mine operations and their mutual relations with the environment will be further developed.

The lowest share was found in publications on the risks associated with the operation of mining machinery. In addition, only three research areas can be distinguished in this group, which relate to the same method of analysis/risk assessment and to issues of maintenance safety and reliability. It should be noted, however, that the last two groups have a total of three articles. Following this, it can therefore be concluded that this is an important research gap identified for the risk area under consideration. This is also confirmed by the mind map that has been prepared, which is shown in Figure 8. However, in the available literature, one may find many publications focused on risk-based maintenance, reliability-based maintenance, failure-based maintenance, or safety engineering issues. Indeed, a broad analysis of resources during selection process performance based on the use of other keywords (e.g., risk-based maintenance, reliability-based maintenance, failure-based maintenance) allowed the authors to find publications on this subject for the mining sector as well. For the taken assumption in the selection process, the additional searching procedure gave the possibility to identify additional 39 papers that address the risk issues in maintenance safety and reliability. Moreover, the evolving concept of Industry 4.0 may lead to the focus of risk analysis and risk management researchers in the forthcoming period on issues related to risks arising from the operation of machinery.

Based on the research gap identified above, there can be identified further research work that can significantly affect the development of this area of science. Mining machines, their reliability and operational safety are the main risk factor occurring in mining processes. Their priority importance can be demonstrated by the fact that most of the standards and directives identified for the mining industry refer precisely to machinery (see Figure 2). This hypothesis is also confirmed by research conducted in analogous human engineering systems, such as production systems. The importance of analyzing the risks associated with machines operation can also be demonstrated by the intensive development of research in the area of risk-based maintenance. The main goal of this concept is to reduce the overall risk that may result as the consequence of unexpected failures of operating facilities [127]. Research on the implementation of this strategy in the field of mining machines maintenance is an interesting area for continuing further research analysis.

The last important research area which the authors would like to mention is financial and economic risk assessment issues.

Although financial and economic risk assessment is a broad research area for many sectors, the authors have not identified any literature related to these issues in the investigated databases during the selection procedure performance. The adopted assumptions (e.g., defined time period restriction, keywords used) have not allowed for finding any relevant publication focused on risk analysis, assessment or management in the mining sector, taking into account financial or economic requirements. The additional analysis of resources during selection process performance based on the use of other keywords (e.g., financial risk, currency risk, economic) gives the authors the possibility to find publications on this subject for the mining sector. However, the vast majority of the publications found in this area relate to the period 1990–2010, and therefore fall outside the assumed time frame of the analysis adopted in the carried out review.

The analysis of the publications also indicates changing trends in the area of risk research conducted in the mining sector. The turn of the 21st century was a period when research on financial and economic risks was in the interest of researchers and practitioners. The last decade has focused researchers' attention primarily on environmental issues. Simultaneously, in the future, the main direction of research development will be the issues related to the operation and maintenance of mines' infrastructure and machinery.

## 6. Conclusions

The prepared systematic literature review was aimed at providing a general overview of related research to risk in mining sector. After a search process that yielded 736 results, 94 papers were selected. These papers were strictly related to the area of analysis, assessment and risk management in mining. They were published in 45 journals, most of which were thematically related to environmental and human health protection. Most of the selected studies were published in the last 5 years, which proves that this research direction is only in its growth phase and should develop in the coming years.

Furthermore, the presented literature review can be a valuable resource for understanding the latest developments in risk management and assessment in the mining sector. Thus, it will be useful to many people including managers, engineers, and researchers, who are interested in risk management/engineering issues. The authors believe that the conducted literature review will introduce the readers to the major up-to-date theory and practice in risk management/assessment problems in the chosen industry sector. The presented study allowed us to identify the thematic structure related to risk assessment/management for the mining sector. In addition, it showed which topics from the studied scientific area dominate in a given country. At the same time, the conducted analysis gave an opportunity to develop future research directions in the areas identified as research and knowledge gaps.

Moreover, when analyzing the risk management/assessment issues in the mining sector, one cannot forget about the possible different risk factors and aspects occurring at various stages of the mining life cycle (MLC). MLC comprises six phases: exploration and feasibility, design and planning, construction and installation, exploitation and mineral processing, mine closure, and post-mining land use. The selected papers on risk management and assessment in mining industry mostly referred to the three last stages of MLC. However, the current literature study (selected 94 papers) did not allow us to perform a full analysis of the significance of risks and their types in the different phases of the MLC. At the same time, preliminary research done by the authors confirms the significance of this issue. Therefore, the authors consider the performance of a review of studies on risk management/assessment in mining life cycle to be the basic direction of their future research studies.

**Author Contributions:** Conceptualization, A.T. and S.W.-W.; methodology, A.T. and S.W.-W.; formal analysis, A.T. and S.W.-W.; resources, A.T., S.W.-W., and A.W.; data curation, S.W.-W.; writing—original draft preparation, A.T., S.W.-W. and A.W.; writing—review and editing, A.T., S.W.-W. and A.W.; visualization, A.T. and A.W.; All authors have read and agreed to the published version of the manuscript.

**Funding:** This research has received funding from European Institute of Innovation and Technology (EIT), a body of the European Union, under the Horizon 2020, the EU Framework Programme for Research and Innovation. This work is supported by EIT Raw Materials GmbH under the Framework Partnership Agreement No. 19036 (SAFEME4MINE. Preventive Maintenance system on safety devices of Mining Machinery). The funders had no role in the design of the study, as well as in the collection, analyses, interpretation of data, in the writing of the manuscript, and in the decision to publish the results.

**Conflicts of Interest:** The authors declare no conflict of interest.

## Appendix A

**Table A1.** Summary of the main standards regarding risk in mining industry.

| Name | Scope | Group |
|---|---|---|
| **PN-N-18001/OHSAS 18001** Occupational health and safety management systems-Specification | Defines the requirements for the occupational health and safety management system. These requirements enable the organization to control occupational risk and improve health and safety. | General |
| **PN-N-18002:2011** Occupational Health and Safety Management Systems. General guidelines for occupational risk assessment | Defines general guidelines occupational risk assessment at workplaces. | General |
| **ISO 31010:2009** Risk management-Risk assessment techniques | Defines guidance on selection and application of systematic techniques for risk assessment | General |

**Table A1.** *Cont.*

| Name | Scope | Group |
|---|---|---|
| **Directive 89/391/EEC** measures to improve the safety and health of workers at work | Defines measures to improve the health and safety of people at work. It sets out obligations for both employers and employees to reduce accidents and occupational disease in the workplace. | Human |
| **Directive 2006/42/EC** on machinery, and amending Directive 95/16/EC (recast) | Application to machinery, interchangeable equipment, safety components, lifting accessories, chains, ropes and webbing, removable mechanical transmission devices and partly completed machinery. | Machines |
| **ISO 12100:2010** Safety of machinery-General principles for design-Risk assessment and risk reduction | Defines basic terminology, principles, and a methodology for achieving safety in the design of machinery. | Machines |
| **ISO 13849-1:2015** Safety of machinery-Safety-related parts of control systems-Part 1: General principles for design | Defines safety requirements and guidance on the principles for the design and integration of safety-related parts of control systems (SRP/CS), including the design of software. | Machines |
| **ISO 13849-2:2012** Safety of machinery-Safety-related parts of control systems-Part 2: Validation | Defines the procedures and conditions to be followed for the validation by analysis and testing of the specified safety functions, the category achieved, and the performance level achieved by the safety-related parts of a control system (SRP/CS) designed in accordance with ISO 13849-1. | Machines |
| **ISO 14121-1:2007** Safety of machinery-Risk assessment-Part 1: Principles | Defines general principles intended to be used to meet the risk reduction objectives established in ISO 12100-1:2003, Clause 5. | Machines |
| **PN-EN 12111: 2014-07** Tunneling machinery-Road headers and continuous mining machines-Safety requirements | Identifies all significant hazards, hazardous situations and events related to road headers and continuous mining machines, used as intended, as well as in the conditions of incorrect use, foreseeable by the manufacturer. | Machines |
| **PN-EN 14973: 2016-01** Conveyor belts used in underground excavations-Electrical and fire safety requirements | Defines the electrical and fire safety requirements for conveyor belts intended for use in underground excavations, in a flammable or non-flammable atmosphere. | Machines |
| **PN-EN 60204-1: 2018-12** Safety of machinery-Electrical equipment of machines-Part 1: General requirements | Applies to electrical, electronic, and programmable electronic equipment and systems for machines not held in hands while working, including a group of machines working together in a coordinated manner. | Machines |
| **PN-EN 62061: 200** Safety of machinery-Functional safety of electrical, electronic, and electronic programmable safety control systems | Defines requirements and makes recommendations for the design, completion, and validation of electrical, electronic, and programmable electronic control systems (SRECS) for machines. | Machines |
| **Directive 94/9/EC -** the approximation of the laws of the Member States concerning equipment and protective systems intended for use in potentially explosive atmospheres | Applies to equipment and protective systems intended for use in potentially explosive atmospheres. | Environment |
| **PN-EN 14591-1: 2006** Explosion protection in underground mine headings-Protective systems-Part 1: Explosion-proof ventilation dam with strength 2 bar | Defines safety requirements for ventilation structures such as dams and explosion barriers, resistant to explosions with a pressure of up to 2 bar. | Environment |
| **PN-EN 14983: 2009** Explosion prevention and explosion protection in underground mining plants-Equipment and protective systems intended for methane drainage | Defines requirements for devices intended for the drainage of underground mining plants. These devices include: fans, compressors and other types of equipment designed to maintain safety. | Environment |
| **PN-EN 60079-0: 2013-03** Explosive atmospheres-Part 0: Equipment-Basic requirements | Defines the basic requirements for the construction, testing and marking of electrical equipment and Ex components intended for use in explosive atmospheres | Environment |
| **PN-EN 1710 + A1: 2008** Equipment and components intended for use in potentially explosive atmospheres in underground mine headings | Defines the requirements for the design of equipment and components used as individual devices or machine parts in underground mine headings endangered by methane and/or coal dust explosion. | Environment |
| **PN-EN ISO 14001** Environmental management systems-Requirements and guidelines for use | Defines the requirements for an environmental management system that an organization can use to improve the environmental effects of its operations. | Environment |

## Appendix B

**Table A2.** Summary of the reviewed papers.

| No. | References | Journal | Publication Year | The Main Target of the Study | Keywords | Search Group | Thematic Group |
|---|---|---|---|---|---|---|---|
| 1 | [40] | Environmental Geochemistry and Health | 2019 | Evaluation of the pollution status and ecological risk of agricultural soils using complex quality indices and new index for ecological risk | Surface soil, trace metal, ecological risk assessment, source identification, geographic information system | Environmental risk assessment | Environment |
| 2 | [54] | Environmental Earth Sciences | 2018 | Assessment of health risk of workers and residents in the vicinity of the active beneficiation of Eshidiya phosphate mine water following EPA risk assessment guidelines (heavy metal contamination levels of mine water) | Phosphate beneficiation, mine water, health risk assessment, heavy metals, carcinogenic and non-carcinogenic risk, Jordan | Environmental risk assessment | Environment |
| 3 | [55] | Human and Ecological Risk Assessment | 2017 | To examine association between perceived environmental exposures from mining activities and subjective health in Northern Ghana using the Upper West Region as a case study | Ghana, Upper West Region, self-rated health, impacted, affected | Environmental risk assessment | Environment |
| 4 | [56] | Environmental Monitoring and Assessment | 2016 | Human health risk assessments for heavy metals contamination around Obuasi gold mine in Ghana | Heavy metals, contamination, gold mining, health risk assessment, Obuasi, Ghana | Environmental risk assessment | Environment |
| 5 | [57] | Bulletin of Environmental Contamination and Toxicology | 2013 | Determining metal pollution (Cu, Zn, Pb, Cd, As, Hg, Ni and Al) and ecological risk in the sediments around Rize Harbor | Metal contamination SQGs, enrichment factor, factor analysis, toxic units | Environmental risk assessment | Environment |
| 6 | [58] | Human and Ecological Risk Assessment | 2018 | Assessment the potential health risk for children and adults connected with concentrations of REEs (Rare Earth Elements) in PM10 | rare earth elements, particulate matter, health risk assessment, Nandan County | Environmental risk assessment | Environment |
| 7 | [59] | Environmental Geochemistry and Health | 2016 | Highlighting the environmental pollution problems and public health concerns of coal mining, particularly the potential occupational health hazards of coal miners exposed in Heshan | Coal mining, polycyclic aromatic hydrocarbons (PAHs), principal component analysis (PCA), incremental lifetime risk (ICLR), Heshan, Guangxi Autonomous Region | Environmental risk assessment | Environment |
| 8 | [60] | Environmental Science and Pollution Research | 2018 | Investigation of the potential harmful element (PHE) concentrations in coal dust and evaluation of the human risk assessment and health effects near coal mining areas | Coal dust, chronic daily intake, chronic risk, cancer risk, coal mines | Environmental risk assessment | Environment |

**Table A2.** *Cont.*

| No. | References | Journal | Publication Year | The Main Target of the Study | Keywords | Search Group | Thematic Group |
|---|---|---|---|---|---|---|---|
| 9 | [61] | Environmental Geochemistry and Health | 2019 | A model specialized to the HRA of abandoned metal mine areas was developed via modification of the Korean guidelines in terms of exposure pathways in the scenario and equations and parameters for estimating human risk | human risk assessment, abandoned mine, heavy metal contamination, carcinogenic risk, non-carcinogenic risk, remediation level | Environmental risk assessment | Environment |
| 10 | [62] | Food Security | 2015 | Measurement of the concentrations of Pb, Cd, Cu, and Zn in paddy soils and rice grains collected from sites close to seven mines in Hunan province and estimation of the degree of paddy soil pollution and human health risks through white rice consumption of these elements around those areas | Heavy metal, potential health risk, mining-affected area, paddy soil, white rice, Hunan province | Environmental risk assessment | Environment |
| 11 | [63] | Journal of Geographical Sciences | 2015 | Development of quantitative estimation of the non-carcinogenic and carcinogenic risks of heavy metals in road dust to local residents of Bayan Obo Mining Region in Inner Mongolia, North China | road dust, heavy metal elements, contamination assessment, health risk assessment, Bayan Obo Mining Region | Environmental risk assessment | Environment |
| 12 | [64] | Environmental Geochemistry and Health | 2013 | Assessment of health risk of mercury pollution via oral exposure to inhabitants in southwestern China | Mercury, heavy metals exposure pathway, average daily intake dose, mining activity | Environmental risk assessment | Environment |
| 13 | [65] | Human and Ecological Risk Assessment | 2017 | Exploration of the distribution characteristics and source of Hg in indoor and outdoor dust of Huainan city; analysis of influencing factors for the Hg concentrations in different districts; Evaluation of the potential risks to adults and children | Mercury, contamination characterization, indoor and outdoor dust, health risk, Huainan | Environmental risk assessment | Environment |
| 14 | [66] | Bulletin of Environmental Contamination and Toxicology | 2019 | Analysis of the degree of pollution by heavy and radioactive metals of the oldest known extraction sites (e.g., Ciudanovita, Lisava, Anina, and Moldova Noua) | Heavy metals, tailings dumps, dose rate, SEM, FTIR, ICP–MS | Environmental risk assessment | Environment |
| 15 | [67] | Environmental Science and Pollution Research | 2016 | The potential chronic risks associated with the exposure to individual and multiple heavy metals by contaminated food consumption were evaluated by calculating the DIR | Heavy metals, contaminated food, heavy metal ingestion, health risk assessment | Environmental risk assessment | Environment |

**Table A2.** *Cont.*

| No. | References | Journal | Publication Year | The Main Target of the Study | Keywords | Search Group | Thematic Group |
|---|---|---|---|---|---|---|---|
| 16 | [68] | Environmental Geochemistry and Health | 2015 | A model of atmospheric particle dispersion from mine-waste dumps to delimit the risk zones of trace element contamination of soil and to determine its influence on the population | Environmental impact, mining activity, potentially toxic elements, soil pollution modeling | Environmental risk assessment | Environment |
| 17 | [69] | Environmental Monitoring and Assessment | 2019 | Assessment of ecological and health risk connected with heavy metals contamination (As, Cd, Cr, Cu, Pb, and Zn) of surface sediments | Gold mining, sediment, trace metals, assessment | Environmental risk assessment | Environment |
| 18 | [70] | Environmental Geochemistry and Health | 2018 | Determining the levels of target PAHs in sediment samples at Okobo-Enjema mine vicinity, and assessment of the potential health risk to benthic organisms and humans on exposure to the PAHs through the sediments | Analysis, concentrations PAHs, risk assessment, source apportionment, sediments | Environmental risk assessment | Environment |
| 19 | [71] | Archives of Environmental Contamination and Toxicology | 2015 | Sampling sites for household dust collection were selected according to building distribution location, building styles, and indoor activities in the Vicinity of Phosphorus Mining, Guizhou Province, China | Not specified | Environmental risk assessment | Environment |
| 20 | [72] | Environmental Geochemistry and Health | 2012 | Evaluating As bio accessibility in stratified samples from a gold mining area and assessing children exposure to As- contaminated materials | Trace elements, in vitro tests, human health, anthropogenic impacts, environmental contamination | Environmental risk assessment | Environment |
| 21 | [73] | Arabian Journal of Geosciences | 2018 | The determination of spatial variation of Zn, Cu, Cr, Pb, Hg, and As in the Junggar coal mine area and risk assessment of soil toxic metals | Coal mining, toxic metal pollution, pollution index (PI), risk assessment, spatial distribution | Environmental risk assessment | Environment |
| 22 | [74] | Environmental Geochemistry and Health | 2017 | Health risk assessment through consumption of vegetables from polluted areas | Mining activity, heavy metals/metalloids, vegetables exposure, hazard | Environmental risk assessment | Environment |
| 23 | [75] | Environmental Earth Sciences | 2018 | Ecological risk assessment for heavy metals contamination (Cu, Cd, Pb, Cr, and Zn) in topsoil of Yedidalga mine harbor in Northern Cyprus | Heavy metals, soil contamination, pollution assessment, ecological risk, Yedidalga mine harbor | Environmental risk assessment | Environment |
| 24 | [76] | Environmental Monitoring and Assessment | 2015 | The concentrations of 12 metals (As, Be, Bi, Cd, Co, Cr, Cu, Hg, Ni, Pb, Sb, and Zn) in soil, agricultural product, and hairy vetch samples were determined to identify relationships between the heavy metal concentrations in the soil and the sources of the heavy metal pollution | Heavy metal, multivariate analysis. Agricultural soil, agricultural products, mining and smelting areas, risk assessment, Enrichment factor, ecological risk | Environmental risk assessment | Environment |

**Table A2.** *Cont.*

| No. | References | Journal | Publication Year | The Main Target of the Study | Keywords | Search Group | Thematic Group |
|-----|-----------|---------|------------------|------------------------------|----------|--------------|----------------|
| 25 | [77] | Environmental Monitoring and Assessment | 2018 | To examine the degree and extent of Pb, Zn, Cd, As, Cr, Cu, Hg, and Ni contamination in soils associated with Jinding mining activities, to assess the potential environmental risk; and to determine the major sources of heavy metal contamination in soils | Heavy metals, soil contamination, pollution indexes, environmental risk assessment, multivariate analysis, spatial distribution | Environmental risk assessment | Environment |
| 26 | [78] | Human and Ecological Risk Assessment | 2018 | Examination of the contents of Cd, Cu, Zn, and As in soil to analyze the efficiency of biochar treatments on bioavailability and speciation distribution of heavy metals in coal-contaminated soil | Heavy metals, biochar, contaminated soil, speciation, bioavailability | Environmental risk assessment | Environment |
| 27 | [80] | Environmental Geochemistry and Health | 2019 | Analysis of soil samples around pristine and major gold-mining areas in Ghana for heavy metals as part of a larger soil contamination and metal background study | Heavy metal, mining, Pristine soil, contamination indices, health risk assessment | Environmental risk assessment | Environment |
| 28 | [81] | Chemistry Central Journal | 2011 | Measurement of the levels of heavy metals (Fe, Mn, Zn, Cu, Ni, Cd and Pb) found in common vegetables grown in contaminated mining areas compared with those grown in reference clear area and to determine their potential detrimental effects | Not specified | Environmental risk assessment | Environment |
| 29 | [82] | Environmental Monitoring and Assessment | 2013 | Analysis of the total contents of selected metals (Ti, Mn, Cr, Pb, Zn, Ni, Cu, As, Hg and Cd) in mine soil around the Lake, and determining the distribution of the metals in the area surrounding the Miyun Reservoir and assessment of heavy metal eco-logical risk in soil using the Igeo index | Heavy metals, soils, sequential extraction, multivariate analysis, risk assessment | Environmental risk assessment | Environment |
| 30 | [83] | Human and Ecological Risk Assessment | 2019 | Application of engineering methods and crop rotation systems to remediate heavy metal contaminated agricultural soil around mining area through a 1-year field experiments | Heavy metal, in situ remediation, gold mining, contaminated agricultural soil | Environmental risk assessment | Environment |
| 31 | [84] | Human and Ecological Risk Assessment | 2017 | Determining the levels of As in 23 vegetable species planted on As-polluted soils and assessing the human health risks of vegetable consumption in the contaminated area | Arsenic, vegetable, soil contamination, mining area, health risk assessment | Environmental risk assessment | Environment |

**Table A2.** *Cont.*

| No. | References | Journal | Publication Year | The Main Target of the Study | Keywords | Search Group | Thematic Group |
|---|---|---|---|---|---|---|---|
| 32 | [85] | Environmental Monitoring and Assessment | 2015 | Assessment of potential ecological risk of heavy metals (As, Cd, Cr, Cu, Hg, Ni, Pb, and Zn) based on the examined data from Hunan Province | Soil heavy metals, potential ecological risk, zinc-lead mining area | Environmental risk assessment | Environment |
| 33 | [87] | Biological Trace Element Research | 2019 | Assessment of dietary exposure to potentially toxic trace elements, particularly Cu, Mo, Hg, Cd, As, and Pb through the intake of selected vegetables grown under the impact of Kajaran' s mining complex | Mining, transfer factor, trace element, risk | Environmental risk assessment | Environment |
| 34 | [88] | Journal of Environmental Management | 2016 | Assessment of long-term effects of anthropogenic pressure of the sulfur industry on turf-covered soils located in the vicinity of the sulfur mine Grzybow, Poland | Sulfur mine, contamination, heavy metal, soil, Poland | Environmental risk assessment | Environment |
| 35 | [89] | Environmental Forensics | 2017 | Trace element analysis and risk assessment in mining area soils from Zhexi river plain, Zhejiang, China | Trace elements, source identification, risk assessment, multiway principal component analysis, river source | Environmental risk assessment, risk analysis | Environment |
| 36 | [90] | Human and Ecological Risk Assessment | 2016 | Assessment of the potential ecological risk (PER) and human health risk of heavy metals (As, Hg, Pb, Cd, Cr and Cu) pollution in urban soils of a coal mining city in East China (Tianjia' an and Datong district) | Heavy metal, mining city, urban soil, spatial distribution, potential ecological risk, human health risk | Environmental risk assessment | Environment |
| 37 | [92] | Environmental Monitoring and Assessment | 2012 | Risk assessment due to intake of heavy metals (pollution of water resources) for East Singhbhum region, India | Groundwater, hazard quotient, heavy metals, risk assessment, uranium mining, iron, manganese | Environmental risk assessment | Environment |
| 38 | [93] | Environmental Monitoring and Assessment | 2018 | Investigation of water body contamination by heavy metals in the vicinity of gold mines and providing of ecological and human health risk assessment estimation | Metal contamination, toxicity, health risk, physicochemical, surface water, mining | Environmental risk assessment | Environment |
| 39 | [94] | Journal of Soils and Sediments | 2018 | Evaluation of the possible risks posed by heavy metals in sediments and pore water | Distribution, heavy metals, particle size, speciation, Yongding River | Environmental risk assessment | Environment |
| 40 | [95] | Environmental Science and Pollution Research | 2018 | The distribution and accumulation of 16 elements (including heavy metals, macro-elements, and other trace elements) in four fish species from Xiang River were investigated through statistical analysis; human dietary exposure to trace elements through fish consumption was evaluated | Bioaccumulation, heavy metal, correlation analysis, Principle component analysis, target hazard quotient | Environmental risk assessment | Environment |

**Table A2.** *Cont.*

| No. | References | Journal | Publication Year | The Main Target of the Study | Keywords | Search Group | Thematic Group |
|---|---|---|---|---|---|---|---|
| 41 | [96] | Human and Ecological Risk Assessment | 2016 | Evaluation of the degree of residents' exposure to seven heavy metals and assessment of the pollution in water and sediment samples at four different locations of the Yellow River, China | Exposure level, hair, heavy metals, risk assessment, Yellow River | Environmental risk assessment | Environment |
| 42 | [97] | Frontiers of Environmental Science and Engineering | 2015 | Distribution characteristics of heavy metal in the groundwater of Chenzhou City, China region (Shizhuyuan Polymetallic Mine) and evaluation of the potential risks to human health as a drinking water source | Groundwater, heavy metal, health risk assessment, mine area | Environmental risk assessment | Environment |
| 43 | [100] | Human and Ecological Risk Assessment | 2016 | Assessment of environmental risk of GolGohar Iron Ore Complex Sirjan | environmental risk assessment, GolGohar Sirjan Ore Complex, wildlife habitat, FMEA | Environmental risk assessment | Environment |
| 44 | [102] | Environmental Technology and Innovation | 2015 | Methodology to conduct risk assessment under uncertainty during a pre-mining phase, where hydrogeological information that characterizes a mine site is limited | Risk analysis, fugacity model, acid rock drainage, uncertainty, probability bounds analysis | Environmental risk assessment | Environment |
| 45 | [103] | Environmental Earth Sciences | 2012 | Environmental risk assessment for acid mine drainage (Zn, Cu, Ni, As, Co, Sb, SO42-, pH, alkalinity) | Acid neutralizing capacity, sulphide metal leaching, Tarkwa Prestea, mine spoil | Environmental risk assessment | Environment |
| 46 | [105] | Human and Ecological Risk Assessment | 2017 | Ecological risk assessment of heavy metals Zhexi river plain, Zhejiang, China using the modified potential ecological risk index (MRI) | mine dumping site, acidification, heavy metals, water soluble, ecological risk | Environmental risk assessment | Environment |
| 47 | [107] | Environmental Monitoring and Assessment | 2019 | Assessment and verification of bio accessibility of Pb and Zn using three different methodologies of sequential extraction and a bio accessibility method; Contamination assessment | Potentially toxic metals, sequential extraction, bio accessibility, potential ecological risk, risk assessment code | Environmental risk assessment | Environment |
| 48 | [108] | Environmental Science and Pollution Research | 2013 | A method for quantifying the impact of mining activities, taking account of the quality of environmental media in the Rosia Montana area | Environmental pollution, impact assessment, risk assessment, mining, Rosia Montana | Environmental risk assessment | Environment |
| 49 | [116] | International Journal of Mining, Reclamation and Environment | 2011 | A due diligence approach to capture and rank the core social issues that impact both the risk of mine feasibility studies in Canada and stakeholder interests regarding mineral resource development within a community's economic sphere of influence | social risk, due diligence, mining communities, impact-benefits agreements, community relations | Environmental risk assessment | Human |

**Table A2.** *Cont.*

| No. | References | Journal | Publication Year | The Main Target of the Study | Keywords | Search Group | Thematic Group |
|---|---|---|---|---|---|---|---|
| 50 | [38] | Mining Technology | 2017 | Method developed to investigate the robustness of stochastic simulations in risk quantification and stochastic optimization studies for a given mineral deposit under operation | Geostatistical simulation, mine reconciliation, stochastic optimization, risk analysis, mining scheduling | Risk analysis | General |
| 51 | [86] | Bulletin of Environmental Contamination and Toxicology | 2015 | Evaluation of ecological impacts of metals in soil from the restored Panyi coal mining area in China | Avian, ecological impacts, intervention, mammalian, soil contamination, plants | Risk analysis | Environment |
| 52 | [119] | Stochastic Environmental Research and Risk Assessment | 2018 | Approach that integrates multi-goal FCM and sensitivity analysis in multi-criteria decision-making process to prioritize and assess the risk associated with working accidents in underground collieries | Workplace accident risk, multi-goal FCM, TOPSIS, sensitivity analysis, underground collieries | Risk analysis | General/Human |
| 53 | [121] | Berg Huettenmaenn Monatsh | 2019 | Comparison of advantages and disadvantages of haulage methods to guarantee optimum choice in terms of evaluation and risk analysis of open-pit mining operations | Bucket wheel excavator, cutting resistance, risk analysis | Risk analysis | Machines |
| 54 | [36] | Iranian Journal of Science and Technology-Transactions of Civil Engineering | 2018 | Risk analysis and risk management of wastewater transmission and treatment including risk categorization, risk reduction and confrontation strategies | Risk assessment, wastewater systems, vulnerability, FAHP, FSAW | Risk assessment | General |
| 55 | [41] | International Journal of Mining, Reclamation and Environment | 2010 | An application of a risk assessment approach in characterizing the risks associated with safety violations in underground bituminous mines in Pennsylvania using the Mine Safety and Health Administration (MSHA) citation database | Safety, standard citations, risk assessment, coal mines, Pennsylvania | Risk assessment | General |
| 56 | [42] | International Journal of Mining, Reclamation and Environment | 2017 | A practical method for assessing the risk of roof falls in coal mines | Underground coal mine, longwall, stability of the roof, roof fall, risk assessment. | Risk assessment | General |
| 57 | [44] | Journal of Safety Research | 2019 | Pythagorean fuzzy numbers-based (PFVIKOR) approach for improving overall safety levels of underground mining by considering and advising on the potential hazards of risk management | Occupational hazards, risk assessment, underground copper and zinc mine, Pythagorean fuzzy set, VIKOR | Risk assessment | General |
| 58 | [46] | Stochastic Environmental Research and Risk Assessment | 2018 | A hybrid information fusion approach that integrates the cloud model and the D–S evidence theory to perceiving safety risks using sensor data under uncertainty | Safety analysis, cloud model, D–S evidence theory, information fusion, tailings dam, sensor data | Risk assessment | General |

**Table A2.** *Cont.*

| No. | References | Journal | Publication Year | The Main Target of the Study | Keywords | Search Group | Thematic Group |
|---|---|---|---|---|---|---|---|
| 59 | [47] | Arabian Journal of Geosciences | 2018 | A decision model is established to evaluate safety of coal mining above confined aquifer | Multi-criteria decision-making, weighted linear combination, geographic information system, maximizing deviation | Risk assessment | General |
| 60 | [49] | Stochastic Environmental Research and Risk Assessment | 2018 | Concepts and methods for evaluating sequential information gathering schemes in spatial decision situations | Value of information, spatial risk analysis, spatial statistic, sequential information, adaptive testing, Bayesian networks, Gaussian processes | Risk assessment | General |
| 61 | [50] | International Journal of Coal Science and Technology | 2016 | A framework to quantify the risk of kriging-based mining projects due to the geological uncertainties | Open pit mine planning, Geological uncertainty, multivariate conditional simulation, grade/tonnage curves | Risk assessment | General |
| 62 | [51] | Journal of Mining Science | 2015 | Geotechnical risk assessment process to suit the underground mining needs | Underground mine, geotechnical accident, risk prevention, risk assessment scope, risk assessment tools | Risk assessment | Environment |
| 63 | [52] | International Journal of Rock Mechanics and Mining Sciences | 2010 | An approach to building damage risk assessment on mining induced areas | Risk assessment, GIS, mining subsidence, building damage | Risk assessment | General/environment |
| 64 | [53] | Human and Ecological Risk Assessment | 2019 | The integrated system comprising deep mining, coal-gangue dressing, and underground backfilling is proposed, followed by analysis of the type, construction, and protection standards of the buildings. Case study for Tangshan coal mine of the Kailuan Group, China | Risk assessment, surface subsidence, backfill mining, dense buildings, environmental protection | Risk assessment | Environment |
| 65 | [101] | Arabian Journal of Geosciences | 2016 | Exploring the impacts of dilution on open pit mines and examines the major risks associated with dilution | Dilution, open pit mines, risk analysis, management tool | Risk assessment | General |
| 66 | [104] | Human and Ecological Risk Assessment | 2012 | A quantitative ERA (QERA) was undertaken of the Magela Creek floodplain, downstream of the Ranger mine, which encompassed point source mining-related risks and diffuse landscape-scale risks | Ecological risk assessment, Ranger uranium mine, landscape-scale risks | Risk assessment | Environment |
| 67 | [106] | Human and Ecological Risk Assessment | 2018 | Overview of environmental impact of underground coal mine technological units | Coal mine, environmental impact assessment, influential parameters; measurement of parameters, risk assessment | Risk assessment | Environment |
| 68 | [109] | Acta Geophysica | 2017 | A GIS approach to the standard deterministic seismic risk assessment by focusing on the potential losses to population and infrastructure | Acid mine drainage, Johannesburg, seismic risk, vulnerability, GIS | Risk assessment | Environment |

**Table A2.** *Cont.*

| No. | References | Journal | Publication Year | The Main Target of the Study | Keywords | Search Group | Thematic Group |
|---|---|---|---|---|---|---|---|
| 69 | [111] | Arabian Journal of Geosciences | 2018 | A GIS- and AHP-based method of risk assessment of coal-floor water inrush on the no. 11 coal seam | Coal floor, water inrush, analytic hierarchy process (AHP), GIS, vulnerability index method (VIM) | Risk assessment | General |
| 70 | [112] | Arabian Journal of Geosciences | 2019 | A method for evaluating high confined water hazard in coal seam floor | Improved analytic hierarchy process vulnerability index (IAHP-VI) method, vulnerability index model, GIS, geological structure | Risk assessment | General |
| 71 | [113] | Journal of Coal Science and Engineering | 2010 | Risk assessment of Xinhe's water inrush evaluation based on quantification theoretical models | Water inrush, assessment, prediction, quantification theoretical, model | Risk assessment | General |
| 72 | [114] | Water Resources and Industry | 2019 | Testing the ability of three assessment methods to adequately reflect water-related risks of a mining operation based on a case study approach for six copper mines | Copper mining, water risk, risk assessment, environmental performance, accountability, sustainability | Risk assessment | General/environment |
| 73 | [117] | Extractive Industries and Society | 2017 | Presentation of the results of the Infectious Disease Risk Assessment and Management (IDRAM) initiative pilot | Extractive industry, emerging infectious diseases, EIDs, infection and prevention control, IPC | Risk assessment | Human |
| 74 | [118] | International Journal of Mining, Reclamation and Environment | 2017 | Method for radiation risk assessment focused on the need of individual dosimetry of all population of the region, country on the basis of, for example, solid-state individual dosimetry (TLDs) | Radiation, dumps, tailing dams, uranium mines, radiation risk | Risk assessment | Human |
| 75 | [122] | International Journal of Occupational Safety and Ergonomics | 2020 | Method for evaluation of the risks which may be available in mechanized coal mines in Turkey, which based on expert knowledge and engineering judgement in linguistic forms implementation | Fuzzy logic approach, fuzzy inference system, Mamdani algorithm, risk matrix, coal mining | Risk assessment | General/machines |
| 76 | [123] | International Journal of Injury Control and Safety Promotion | 2015 | A method based on the concepts of task and accident mechanisms for an initial risk assessment by taking into consideration the prevalence and severity of the maintenance accidents reported | Risk assessment, accident mechanisms, occupational safety, maintenance, accident analysis | Risk assessment | General |
| 77 | [125] | Transactions of the Institutions of Mining and Metallurgy, Section A: Mining Technology | 2015 | Risk assessment of an exploration drill rig focusing on hazards connected to maintenance and operability of the equipment and identification of safety improvements | Equipment design, drill rig, safety in design | Risk assessment | General |

**Table A2.** *Cont.*

| No. | References | Journal | Publication Year | The Main Target of the Study | Keywords | Search Group | Thematic Group |
|---|---|---|---|---|---|---|---|
| 78 | [45] | International Journal of Injury Control and Safety Promotion | 2017 | Risk-based decision-making methodology proposed for selecting an appropriate safety measure system in relation to an underground coal mining industry with respect to multiple risk criteria | Underground coal mining industry, risk-based decision making, interval-valued fuzzy set theory, fuzzy risk analysis | Risk decision | General |
| 79 | [79] | Journal of Agricultural and Environmental Ethics | 2019 | Determination of the concentrations of heavy metals in the soil around "Larga de Sus" abandoned mine (Zlatna town, Romania) and assessment of the potential ecological risk of heavy metals in soil as a tool to help in the decision-making process | Environmental ethics, ecological risk assessment, heavy metals, soil pollution, sustainability | Risk decision | Environment |
| 80 | [2] | Resources Policy | 2014 | Review of international case studies concerning the functioning and liquidation of the mining enterprises along with an extended analysis of the effects of collieries' liquidation in Poland in the hard coal mining restructuring process | Risk management in hard coal mining industry, collieries' liquidation, corporate social responsibility in hard coal mining | Risk management | General |
| 81 | [34] | Journal of Loss Prevention in the Process Industries | 2013 | New practical approach to risk management in underground goldmines in Quebec | Mining projects, underground goldmines, risk management, occupational health and safety (OHS), multi-criteria analysis (AHP) | Risk management | General |
| 82 | [35] | Procedia Engineering | 2015 | Contribution to create a framework for implementing or improving risk management practices in procurement activities in mining companies by proposing a knowledge-based supporting system | Computer prototype, knowledge-based system, maturity models, procurement and contracting, risk management | Risk management | General |
| 83 | [37] | International Journal of Mining Science and Technology | 2017 | Multivariable function analysis methodology approach based on complex system modelling and through real data corresponding to a risk management tool in the mining sector | Risk, risk management, complex systems, mining, decision making | Risk management | General |
| 84 | [39] | Georisk | 2014 | Introduction of the copulas to mining engineering practitioners | Monte Carlo simulations, mining engineering, value-at-risk, copulas, Spearman's rank, Kendall' s tau | Risk management, assessment | General |

**Table A2.** *Cont.*

| No. | References | Journal | Publication Year | The Main Target of the Study | Keywords | Search Group | Thematic Group |
|-----|-----------|---------|-----------------|----------------------------|----------|-------------|----------------|
| 85 | [43] | Ergonomics | 2017 | Aimed at examining safety leadership through a systems-thinking lens by testing the applicability of a popular systems analysis framework in the safety leadership context | Safety leadership, systems-thinking, safe performance, learning from incidents | Risk management | General |
| 86 | [48] | International Journal of Management Science and Engineering Management | 2019 | Method focused at identifying volatile risk events by integrating the AHP, expert questionnaire survey and sensitivity analysis | Analytic hierarchy process, expert questionnaire survey, sensitivity analysis, risk, mining | Risk management, assessment | General |
| 87 | [91] | Sustainable Water Resources Management | 2019 | Studying priority heavy metals (Ag, Al, As, Cd, Cu, Fe, Mo, Ni, Pb, Sn, Sb, Se, Zn, Hg, Te) in the water of minor rivers in the Voghchi and Meghri basins, surface sources of centralized drinking water supply and drinking water supplied to urban and rural population of the mining region in South Armenia | Drinking water, heavy metals, mine pollution, health risk assessment | Risk management | Environment |
| 88 | [98] | Journal of Cleaner Production | 2016 | Providing mine operators with an organized informational framework that could be applied during future underground coal mine closures independent of the environmental problems faced and connected to the types and characteristics of coal and the exploitation methods used | Sustainability, mine closure, underground coal mining, environmental risks, risk management, management tool | Risk management | General/environment |
| 89 | [99] | Environmental Science and Pollution Research | 2013 | The need for re-assessing the potential of mining in the context of sustainable management of natural capital is discussed and a renewed focus on the role of mining from a systems perspective is proposed | Mining, risk assessment, environmental impacts, sustainable resources management | Risk management, assessment | Environment |
| 90 | [110] | Rock Mechanics and Rock Engineering | 2010 | Describing of a seismic risk management philosophy for underground, hard rock mines, based on the application of simple and practical micro seismic data interpretation and analysis techniques | Mining, rock mechanics, rocks engineering, stress, rock burst, mining inducted seismicity, seismic risk, seismic hazard, mine seismology, failure mechanism | Risk management | General |
| 91 | [115] | Journal of Environmental Management | 2016 | Model to identify critical surface water risk zones for an open cast mining environment, taking Jharia Coalfield, India as the study area | Coal mining, GIS, pollution reduction, remote sensing, risk potential index, surface water pollution | Risk management | Environment |

**Table A2.** *Cont.*

| No. | References | Journal | Publication Year | The Main Target of the Study | Keywords | Search Group | Thematic Group |
|---|---|---|---|---|---|---|---|
| 92 | [120] | Journal of Loss Prevention in the Process Industries | 2019 | Discussion from a near-miss safety management system perspective in terms of methods to foster both risk avoidance and locus of control in an effort to reduce the probability of near misses and lost time at the organizational level within the process industry and other high-hazard industries | Health and safety management system, locus of control, lost time incident, mining, near miss incident, Poisson regression, risk avoidance | Risk management | General/human |
| 93 | [124] | Production Planning and Control | 2018 | Theoretical framework, which describes the most recognized tools to be used in each of the proposed phases of failure mode analysis | Failure mode, maintenance, problem analysis, reliability, risk, TRIZ | Risk management | General/Machines |
| 94 | [126] | Safety Science | 2015 | This article examines the process of industry-wide OHS/safety information management in the Australian coal industry | Knowledge sharing, coal industry, knowledge management, knowledge/DIKW hierarchy, Bow-tie analysis, interactive database | Risk management | General |

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
