# Peer review of "Risk Assessment Methods in Mining Industry—A Systematic Review"

_applsci, doi:10.3390/app10155172_

Round 1
Reviewer 1 Report
The paper is well written and it is a literature review on the risk management and assessment publications on the topic of mining. The results of the study are disturbing, considering the large extent and the high number of operating and/or closed mines worldwide. The fact that only 95 papers have been validated for the last decade in this field is an alarming conclusion that research on this topic is not as frequent as it should be.
In order to understand the importance of the paper’s findings, I suggest that authors make also a review (inventory) of the existing mines (mining companies operating throughout the world), in order to point out the scarcity of the research papers as compared to the total number of the mining facilities in the world. I believe this approach and a short debate on this issue would improve the added value of the paper.
Author Response
We are very thankful to the Reviewer for this/her valuable comments and suggestions to our paper “RISK ASSESSMENT METHODS IN MINING INDUSTRY – A SYSTEMATIC REVIEW”. We have revised the manuscript taking into account these valuable inputs.
The responses to the comments from the Reviewers are submitted in the attachment.

Reviewer 2 Report
After reviewing the manuscript, I think the manuscript has several shortcomings:
- For instance, the title says "Risk assessment methods in mining industry - a systematic review" but the risk assessment methods are not described and discusses. Besides, there are no new interesting information related to the field.
- The information displayed in sections 2 and 3 (Review methodology and Results) can be obtained by just searching an article in Scopus or any other web browser.
- The format of citing the documents is not correct. It starts at the 54 reference
Author Response

(The authors gave the same response as above.)

Reviewer 3 Report
The review article is an interesting compilation of publications from the past 10 years on risk assessment and management in the mining industry. The methodology used to screen the publications is sound, although it unavoidably contains subjective choices. The interpretation of the results is very short and the identified knowledge gaps are restricted to operational considerations. An improved results discussion section is required to make this article more complete, less operation-oriented, and more susceptible to publication. My recommendation is to reject and encourage resubmission once the following comments are addressed.
Line 42: replace “systems” by “system”
Figure 1 and Appendix 1: Identify the countries / regions where these standards are applied. Are they applied worldwide or restricted to European countries?
Table 1: Should be improved. The subjects are not really defined by the titles; add information on specific topics discussed in each year.
Line 132: What are the four inclusion criteria?
Lines 136-137: It is not clear, was the screening process done manually or with the use of the filtering tool in PRIMO?
Figure 2: In caption, correct PRISMA. In the first box on the right side, the number of records do not add up (8819 + 93084 ≠ 106911)
Figure 6: Combine the different spelling of words into a single bar: risk assessment + risk assesment, heavy metals + heavy metal
Figure 6: What is “metals middot”?
Figure 7:
- In “environment”, what is the difference between the box “environmental impact on mines” and “impact of mines on environment”? The first one seems related to risk, while the second on consequences. I suggest rewording the titles in the boxes to be more representative.
- What is the meaning of the different dashed lines between boxes and bubbles?
- Do some publications cover more than one grey box topic? Can lines be drawn between main groups?
- What about economic risks? Where they encountered in the literature review and intentionally screened out?
Lines 292-293: Reword the sentence starting with “The solution based…”
Section 5:
- Add discussion on economic or financial risk assessment
- I don’t agree with the identified research gap. A lot of work is being done on preventive maintenance and machine reliability, and in the context of automation and electrification of machinery. The fact that most standards are related to machines proves that a lot of work has been done and implemented already.
- The risk analysis system (machine, human factor, environment) should be compared not only to the number of publications that passed the authors’ screening, but also their implication on the mining industry’s global risks. Which of the risk aspect is more important at each stage of the mine life cycle? Main risks do not have the same source during exploration, operation, closure and rehabilitation.
- The fact that many publications are related to environmental risk highlights a knowledge gap, and it is expected that environmental risk will be the subject of many more publications in the future.
- The different risks should be associated with the evolution of corporate social responsibility in the mining industry. In the past, operational risks were more commonly addresses, resulting in standards and norms related to machines. More recently, health and safety and environmental concerns emerged, which triggered new research fields. These concerns are not shared equally around the globe, was there any trend in the geographical origin of the different type of risks in publications?
- Need to separate discussion and conclusions.
Appendix B is missing
Author Response
We are very thankful to the Reviewer for his/her valuable comments and suggestions to our paper “RISK ASSESSMENT METHODS IN MINING INDUSTRY – A SYSTEMATIC REVIEW”. We have revised the manuscript taking into account these valuable inputs. The responses to the comments from the Reviewers are submitted in the attachment.
